# Lung Ultrasonography and Clinical Follow-Up Evaluations in Fattening Bulls Affected by Bovine Respiratory Disease (BRD) during the Restocking Period and after Tulathromycin and Ketoprofen Treatment

**DOI:** 10.3390/ani12080994

**Published:** 2022-04-12

**Authors:** Enrico Fiore, Anastasia Lisuzzo, Andrea Beltrame, Barbara Contiero, Matteo Gianesella, Eliana Schiavon, Rossella Tessari, Massimo Morgante, Elisa Mazzotta

**Affiliations:** 1Department of Animal Medicine, Production and Health, University of Padua, Viale dell’Università 16, 35020 Legnaro, Italy; anastasia.lisuzzo@phd.unipd.it (A.L.); barbara.contiero@unipd.it (B.C.); matteo.gianesella@unipd.it (M.G.); rossella.tessari@unipd.it (R.T.); massimo.morgante@unipd.it (M.M.); emazzotta@izsvenezie.it (E.M.); 2Independent Researcher, 1016 Isola Rizza, Italy; andrea.beltrame87@gmail.com; 3Istituto Zooprofilattico Sperimentale delle Venezie (IZSVe), Viale dell’Università 10, 35020 Legnaro, Italy; eschiavon@izsvenezie.it

**Keywords:** TUS, bovine respiratory disease, pulmonary lesions, clinical follow-up

## Abstract

**Simple Summary:**

The restocking period is a crucial time for fattening bulls: stressors such as transport predispose animals to bovine respiratory disease, reducing animal well-being and causing economic losses. Ultrasonography is a non-invasive, real-time, and portable method frequently employed in farming. The aim of this study was to evaluate the feasibility of the thoracic ultrasound in identifying animals with bovine respiratory disease on a restocking farm, and in assessing the efficacy of tulathromycin and ketoprofen treatment in sick animals. Our findings reveal that lung ultrasound is a useful diagnostic tool for screening respiratory diseases. Moreover, tulathromycin and ketoprofen treatment was effective in reducing both clinical symptoms and lung lesions.

**Abstract:**

Bovine respiratory disease (BRD) is a global infectious disease, causing decreased well-being and economic losses in livestock, frequently during the restocking period. The aim of this study was to evaluate the feasibility of thoracic ultrasonography (TUS) to assess BRD in restocked animals, and the effectiveness of tulathromycin and ketoprofen treatment in sick animals. A total of 60 Limousin fattening bulls were enrolled. On the day of restocking (T0), the animals were divided into two groups based on TUS assessment of six lung areas: group C (ultrasonography score (US score) < 3) and group D (US score ≥ 3). Group D received a single administration of tulathromycin and ketoprofen at T0: this group was revaluated after 1.5, 3, 7, and 14 days. Both groups were revaluated after 21 days. The two groups showed a significant difference both in US score and clinical symptoms (respiratory score, nasal and ocular discharges, and rectal temperature) at T0. In group D, the treatment was effective in improving the clinical symptoms and US score, particularly reducing the severity of lung lesions. TUS represents a non-invasive and cost-effective tool for BRD early diagnosis and for monitoring treatment efficacy in restocked livestock.

## 1. Introduction

Bovine respiratory disease (BRD) is a syndrome involving infectious agents, host immune response, and environmental factors. Stressors such as transport may predispose animals to primary infection [1,2,3]. Viral agents such as Bovine Herpesvirus type 1 (BHV-1), Parainfluenza-3 virus (PI3), Bovine Viral Diarrhea Virus (BVDV), Bovine Adenovirus, and Bovine Respiratory Syncytial Virus (BRSV) can cause BRD and increase the susceptibility to secondary bacterial infections generally due to *Mannheimia haemolytica*, *Mycoplasma bovis*, *Pasteurella multocida*, *Histophilus somni*, *Salmonella* spp., or *Actinomyces pyogenes*. In addition, some of these bacteria can also cause a primary infection [3,4,5]. BRD clinical presentation comprises systemic signs and respiratory symptoms [6]. Commonly, the observations of these clinical signs are utilized for BRD diagnosis in-field. However, the heterogeneous signs and the non-specific clinical manifestation frequently do not lead to a defined diagnosis; in fact, the combination of these parameters presents a low sensitivity and specificity. Moreover, subclinical BRD is characterized by the absence of clinical signs and the presence of lung lesions. Misdiagnosis results in the administration of medical treatment to healthy subjects or the missed recognition of sick animals. These situations represent critical points in animal health management and economic losses [6,7,8].

This syndrome is one of the most significant diseases in the beef industry, which may negatively influence the growth, long-term productivity, and costs due to treatment, mortality, and decreased feed efficiency [7,9,10]. Early diagnosis and medical treatment are decisive in improving prognosis and outcome of animals with BRD [8].

Ultrasonography is a non-invasive, cost-effective, and portable method for investigating different structures in real time [11,12]. Regarding BRD, it can be used for diagnosis and can take only 20–30 seconds [13]. The lung ultrasounds present a higher sensitivity (80–94%) and specificity (94–100%) [13] than clinical scoring to detect lung lesions (sensitivity and specificity of 61.8% and 62.8%, respectively) [8,10,14,15]. The increased sensitivity of ultrasonography is important for the early detection of diseased animals, thus reducing animal suffering and duration of disease, whereas the increased specificity is important for the accurate assessment of animal health status to avoid unnecessary antimicrobial treatment, reducing the risk of developing antimicrobial resistance [16]. Medical protocols for the treatment of BRD generally consist of the administration of Gram-negative spectrum antibiotics. Among these, macrolides are used for BRD because of their bacteriostatic capacity, accumulation in leukocytes and bronchial secretions, and high concentration in lung tissue compared to plasma [4,17]. Tulathromycin is a macrolide that has fewer reported morbidity, mortality, and retreatment events than other macrolides (tilmicosin and gamithromycin), tetracyclines (oxytetracycline), amphenicols (florfenicol), and cephalosporin (ceftiofur) [2,17].

The aim of this study was to evaluate the feasibility of thoracic ultrasonography (TUS) in the assessment of fattening bulls with respiratory disease during the restocking period. Furthermore, the response to tulathromycin and ketoprofen treatment in animals with BRD was monitored and analyzed by TUS over 21 days.

## 2. Materials and Methods

### 2.1. Animals and Clinical Examination

This study consisted of a case time-series analysis, conducted from 5 to 26 February 2021, on a single beef-fattening herd located in the province of Verona (Veneto, Italy). A single stock of 60 Limousin fattening bulls was enrolled in this study with an average weight of 364.5 ± 6.42 kg and an age of 10.23 ± 1.37 months. Animals came from France and were kept in outdoor dirt-floor pens. A total mixed ration (TMR) was provided once a day and water was ad libitum. 

All animals were weighed and vaccinated against bovine Parainfluenza-3 Virus (PI3), Bovine Viral Diarrhea Virus (BVDV), Bovine Herpes Virus-1 (BHV-1), and Bovine Respiratory Syncytial Virus (BRSV) (Cattle Master 4, Zoetis Italia S.r.l., Rome, Italy) on the day of restocking and after 21 days. 

The fattening bulls were singularly examined in a cattle crush (Taurus MC Deluxe Gate; Te Pari, Oamaru, New Zealand) where they received a clinical examination by a veterinarian from the Veterinary Teaching Hospital at the University of Padua. The body temperature assessment was performed both with a rectal thermometer and with a thermal imaging camera (ThermaCam P25 Model, Flir Systems, Boston, MA, USA) from the medial canthus of the eye at a distance of 1 m. Furthermore, the presence of respiratory symptoms, such as cough, nasal, and ocular discharges, was evaluated to assign a respiratory score (RS) according to McGuirk and Peek, 2014. According to this scoring system, if the animal presents a score of at least 5, it is considered to be sick. In addition, if the animal has a score of at least 2 in at least two parameters, it is still considered to be a sick animal (percentage of diseased animals according to respiratory score; PDA). Nasal and ocular discharge evaluations were converted to numeric scale: absent = 0; unilateral = 1; bilateral = 2; and abundant bilateral = 3.

### 2.2. Thoracic Ultrasonography Evaluation

The thorax was shaved before performing thoracic ultrasonography (TUS), and ethyl alcohol (90%) was used as a transducing agent. TUS of six lung areas (caudal (10th–7th intercostal space (ICS)), middle (6th–5th ICS), and cranial (4th–3rd ICS) of both lung sides) was performed using a portable ultrasound scanner (MyLabOneTM, Esaote S.p.a., Genova, Italy) equipped with a multi-frequency convex probe (SC3421, Esaote S.p.a., Genova, Italy; 2.5–6.6 MHz). The TUS evaluations were performed using a setting frequency of 4.3 MHz, 15 cm depth acoustics window, 100% grayscale gain, and time-gain compensation in a neutral position. Images were saved in a digital imaging and communications in medicine (DICOM) format and used for post-sampling quantitative assessment (MyLabDeskTM, Esaote S.p.a., Genova, Italy).

The ultrasonography score (US) was attributed according to a 6-point scale (Ollivett and Buczinski, 2016). Lung lesions such as hepatization areas and fluid alveolograms were measured in cm^2^ for the six lung areas. The total lung consolidation was calculated by adding the pulmonary parenchyma hepatization regions and fluid alveolograms of each investigated area.

The lesion score (LS) was calculated for each of the six lung areas by converting the TUS findings into a numeric scale: absence of lesions = 0; comet tail = 1; hepatization = 2; fluid alveolograms = 3; comet tail and hepatization = 4; comet tail and fluid alveolograms = 5; hepatization and fluid alveolograms = 6; and comet tail, hepatization, and fluid alveolograms = 7. The global lesion score (GL) represents the total of the LS of each investigated lung area. 

### 2.3. Experimental Study Design and Laboratory Analysis

On the day of restocking, animals were divided into two groups according to US: group C and group D. Group C or the control/healthy group comprised 29 fattening bulls with an US score of <3. Group D or the disease group comprised 31 fattening bulls with an US score of ≥3. According to the literature (Ollivett and Buczinski, 2016), these animals were considered to be affected by BRD and received a deep nasal swab before the medical treatment (35 cm, equipped with sheath and placed in an agar blister pack; Medical Wire & Equipment Co. Ltd, Corsham, UK). The nasal swabs were refrigerated at 4 °C and transported to the laboratory of IZSVe within 1 hour. These samples were tested for bacterial culture and determination of Minimum Inhibitory Concentration (MIC), according to the validated protocol of the IZSVe. Furthermore, Group D received a single SC injection of a long-acting macrolide and non-steroidal anti-inflammatory (FANS) (tulathromycin and ketoprofen, 2.5 mg/kg + 3 mg/kg, Draxxin plus, Zoetis Italia S.r.l., Rome, Italy). 

The TUS assessments were performed at time 0 (T0; day of restocking) and time 5 (T5; after 21 days) in the healthy (group C) and sick animals (group D). Moreover, group D was evaluated with TUS at time 1 (T1; after 1.5 days), time 2 (T2; after 3 days), time 3 (T3; after 7 days), and time 4 (T4; after 14 days) (Figure 1). 

### 2.4. Statistical Analysis

Statistical analysis was performed using R software v. 4.0.3 (Team R Development Core, 2018) implemented with “rcmdr” package and with SAS system software (version 9.4; SAS Institute Inc., Cary, NC, USA). The normal distribution was assessed by Shapiro–Wilk normality test before all statistical analysis. Statistical significance was set at *p* ≤ 0.05.

The comparisons between group C and D at T0 and T5 were made with unpaired two-samples Wilcoxon test for not normally distributed data (US, RS, nasal and ocular discharges, GL, and LS). The PDA was evaluated with z-test for two samples. The normal distributed data (rectal temperature, thermography temperature, hepatization areas, and fluid alveologram areas) were evaluated with t-test. The total hepatization was evaluated with a linear mixed model (LMM) which considered group, time, and their interaction as fixed factors. Rectal temperature was included as covariate, whereas animal as random and repeated effect. The total fluid alveolograms were evaluated with zero-inflated points generalized linear mixed models (GLIMMIX) which considered the same factors as the previous one. Poisson distribution was assumed, as was a logistic link function.

The follow-up of group D during the time was performed with the Kruskal–Wallis test for non-normally distributed data (US, RS, nasal and ocular discharges, GL, LS, hepatization areas, and fluid alveologram areas). The PDA was evaluated via k proportions chi-squares test. The rectal temperature and thermography temperature were evaluated with an LMM which included the factor of time and the repeated measurements of animals, considered as random. The total hepatization was evaluated with an LMM which considered the factor of time, repeated measurements for the animal, and the covariate of rectal temperature. The total fluid alveolograms were evaluated with a GLIMMIX which considered the same factors as previous models.

For all mixed models, post hoc pairwise comparison among least squares means were performed using Bonferroni correction.

Rectal temperatures and thermography temperatures were standardized for daily values using z-score to reduce the environmental influence between days. A Pearson correlation was used to establish the correlations between the two measurements. 

## 3. Results

Animals affected by BRD (group D) tested positive for *Pasteurella multocida* and *Mannheimia haemolytica* using a deep nasal swab. A single pool collected from three animals showed weak reactivity for *Mycoplasma bovis*. Based on MIC analysis, all these pathogens were sensitive to tulathromycin, the antibiotic selected to treat sick animals.

### 3.1. Difference between the Healthy Group and the Group Affected by BRD on the Day of Restocking and the End of Study 

The two groups showed a significant difference in US (*p* < 0.001), RS (*p* < 0.001), PDA (*p* = 0.001), nasal and ocular discharges (*p* < 0.001 and = 0.04, respectively), and rectal temperature (*p* < 0.001) at T0. However, at T5 exclusively the RS and PDA remained statistically different between the two groups (Table 1). 

A significant difference in GL (*p* < 0.001), total hepatizations (*p* < 0.001), and total alveolograms (*p* = 0.01) was reported between the two groups. Furthermore, at T0, significant differences in TUS evaluation were reported between group C and D: middle left lung (LS (*p* < 0.001), hepatization area (*p* < 0.001), and fluid alveologram area (*p* = 0.02)), cranial left lung (LS (*p* = 0.01), hepatization area (*p* = 0.01), and fluid alveologram area (*p* = 0.01)), middle right lung (LS (*p* = 0.001), hepatization area (*p* < 0.001), and fluid alveologram area (*p* = 0.05)), and cranial right lung (LS (*p* = 0.05), hepatization area (*p* < 0.001), and fluid alveologram area (*p* = 0.02)) at T0. However, only the hepatization area of the middle left lung (*p* = 0.05) and the cranial right lung (*p* = 0.02) remained statistically different between the two groups at T5 (Table 2).

### 3.2. Clinical Follow-Up in animals affected by BRD

Group D showed a significant improvement after the medical treatment for all evaluated parameters (US (*p* < 0.001), RS (*p* < 0.001), PDA (*p* < 0.001), nasal and ocular discharges (both *p* < 0.001), and rectal temperature and thermography temperature (both *p* < 0.001)) (Table 3). 

The US at T0 (day of restocking and treatment) differed significantly from T1, T2, T3, T4, and T5. The RS at T0 differed significantly from T3 and T4. PDA at T0 did not show significant differences during the sampling period. The nasal discharge at T0 differed significantly in T4, while ocular discharges at T0 differed significantly from T2, T3, and T4. Rectal temperature at T0 differed significantly from T1, T2, T3, and T4. Thermography temperature at T0 differed significantly from T0, T1, T2, T3, T4, and T5.

Group D showed a significant improvement in TUS findings during the follow-up: GL (*p* < 0.001), total hepatizations (*p* < 0.001), total alveolograms (*p* < 0.001), middle left lung (LS (*p* < 0.001), hepatization area (*p* < 0.001), and fluid alveologram area (*p* = 0.02)), cranial left lung (LS (*p* = 0.03) and fluid alveologram area (*p* = 0.03)), middle right lung (LS (*p* = 0.001), hepatization area (*p* < 0.001), and fluid alveologram area (*p* = 0.02)), and cranial right lung (hepatization area (*p* = 0.02)) (Table 4). 

The GL at T0 differed significantly from T2, T3, T4, and T5. Total hepatizations at T0 differed significantly from T1, T2, T3, T4, and T5. Total of fluid alveolograms at T0 differed significantly from T2, T3, T4, and T5. The middle areas of both lung sides showed a significant improving trend: the LS of both sides at T0 differed significantly from T3, T4, and T5. The middle left and middle right lung areas reported remarkable variations during the time points: the hepatization areas at T0 differed significantly from T2 and T3 until T5 for both sides. Similarly, the fluid alveologram areas at T0 differed significantly from T3 and T4 until T5 for both sides. The LS of the cranial left lung showed a significant trend over time with significant differences between T0 and T3, T4, and T5. In the same area, the fluid alveolograms’ presence showed a decreasing trend with a difference between T0 and T4–T5. The cranial right lung showed a significant decrease exclusively for hepatization areas, reporting a difference between T0 and T3–T5.

The Pearson correlation between the rectal temperature and thermography temperature converted into z-score was statistically significant (*p* < 0.001) and reported a correlation of r = 0.383. 

## 4. Discussion

Restocking constitutes a crucial point for fattening bulls’ health management; in fact, transport represents a stressor that may negatively influence beef cattle production and health, often resulting in increasing susceptibility to BRD development. A feasible and early identification of these diseases is necessary in the feedlot to avoid economic losses and preserve animal welfare [18]. At present, diagnosis is mainly based on clinical assessment by pen-checkers. However, this method is not accurate (sensitivity and specificity close to 60%): identification and classification of clinical signs of BRD is complicated even when carried out by expert healthcare professionals, due to the varied and confounding presentation of the disease, especially when animals are stressed [19]. 

### 4.1. Difference between the Healthy Group and the Group Affected by BRD on the Day of Restocking and the End of Study 

In this study, clinical parameters such as rectal temperature, cough, and nasal and ocular discharges were used to assign an RS, indicated by McGuirk and Peek (2014). The absolute value of the RS and the PDA showed a greater value in group D compared to group C at T0 and T5. Animals affected by BRD reported a high RS at the beginning of the study, likely due to the respiratory disease. Interestingly, the healthy group reported a PDA of around 40% both at T0 and T5. These results highlight the low specificity of BRD diagnosis merely based on clinical illness [15]. The misclassification of healthy or sick animals in this study may be related to the influence of the animal’s stress on some clinical parameter (cough and rectal temperature). In fact, no animal had a cough during all time points, probably due to a masking behavior/inhibition of the animals related to the operator’s presence [19,20]. Furthermore, a higher rectal temperature may be associated with excitement or stress or overcrowded transport, despite clinical disease [8]. Nevertheless, the RS was originally designed for the classification of calves; the inaccuracy in the detection of respiratory disease in beef fattening bulls is probably due to the greater variety of clinical presentation and severity of BRD in adult animals and their behavioral response to stressors. Nasal and ocular discharges are considered as subjective parameters for respiratory sign assessment, while rectal temperature evaluation is an objective method [8]. Generally, animals affected by BRD show an increased body temperature as the first clinical manifestation followed by nasal and ocular discharges [20,21]. At the beginning of the present study, group C reported no nasal discharge, while group D showed bilateral discharge. Remarkably, at the end of the study (T5), the healthy group showed a worsening of nasal discharge. This finding may be associated with a mild respiratory form, not progressing to systemic or pulmonary disease. In fact, group C showed no hyperthermia, and an US of 2 at T5 which was not associated with a diseased state [13]. Ocular discharge was absent in both groups at T0 and T5, although the interquartile range reported greater variability in group D than in group C at T0; this finding may be associated with the clinical evolution of the disease. Clinical signs may vary in intensity and duration, and mild signs are associated with viral pathogens [3,6]. According to our results, in the case of BRD, it is possible to find respiratory symptoms referring to the upper respiratory tract (nasal and ocular discharges) and not to the deep respiratory tract (lungs and pleura), or vice versa. In addition, sick animals may not show clinical signs indicative of BRD at all [8]. 

Generally, clinical examination is unable to detect the early stages of BRD, whereas increased body temperature is often one of the first alterations observed during the disease [21,22]. Abutarbush et al. (2012) reported that rectal temperature ≥ 40 °C defines fever, which is the most common finding in clinical BRD [14]. Accordingly, in the present study, rectal temperature was greater in group D than in group C at T0. However, no difference was found between the groups at T5. Thermography is a non-invasive method which can detect small changes in body temperature [23,24]. It has been successfully used to detect an early rise in temperature in animals with BRD. In particular, the thermographic image of the medial canthus of the eye detects changes in temperature earlier than the nose or ear [7,22,25]. In this study, a thermographic evaluation of the medial canthus of the eye showed no differences between the groups at the beginning and end of the study. However, environmental factors should be considered in this type of analysis, such as sunlight, humidity, wind, dirty surface, environmental temperature, exercise, and stress [22,23]. A non-standardized condition due to weather conditions and animal management may have interfered with our results. These observations may explain the low correlation (38.3%) between rectal temperature and thermography temperature. 

The TUS of a normal lung represents a continuum between the costal and pulmonary pleura: these structures are not distinguished in normal subjects, and they are represented by a smooth hyperechoic line between the lung and thoracic wall muscles [26]. Below the hyperechoic line of the pleura, the lung shows the characteristic of well-ventilated tissue with the presence of a reverberation artifact [27]. This artifact is characterized by multiple-echo ultrasonic pulse bouncing between two or more highly reflective structures within the axis of the sound beam [28]. Comet-tail artifacts can be observed from the pleura to the deepest part of the scan site, and it is the most common ultrasound abnormality of lung tissue that can also occasionally be observed in healthy subjects [27,29]. Eventually, this artifact radiates from non-aerated areas due to a small accumulation of exudate, blood, mucus, or oedema fluid, and creates multiple reflections [26]. In fact, comet-tail artifacts are characterized by the presence of a thin bundle of strictly hyperechoic lines [28]. The bacterial and viral agent of BRD may result in an area of non-aerated lung lobules that alters ultrasound images of the lung: from reflective tissue with reverberation artifacts to a hypoechoic or anechoic structure. The hypoechoic zones with a liver-like texture are called hepatization [26,30], whereas anechoic zones with comet-tail artifacts may represent fluid-filled alveoli. In severe and chronic stages of lung disease, echogenic bands can be observed within hypoechoic fluid, and they may represent deposits of fibrin [26,27].

In this study, groups differed for the LS in the middle and cranial areas of both lung sides at T0. The healthy group did not report any changes in the lung ultrasound; on the contrary, in animals with BRD, these areas of the lungs were affected by hepatization and fluid alveolograms, or combinations of these or with comet tails in group D. These findings confirm that these areas were the most affected during BRD [27,30]. Moreover, the higher GL, total hepatization areas, and total fluid alveologram areas were reported in group D at T0 and described a severe lung disease and possibly systemic pathology in BRD-affected animals. As mentioned above, due to the difficulty of clinical evaluation, severe lung lesions might not have been identified early if the TUS technique had not been used. 

At the end of the study, LS, GL, and size of lesion areas, for both hepatization and fluid alveolograms, showed no difference between groups, except in the size of hepatization in the middle left and right cranial lungs. In diseased animals, the administration of medical treatment through early detection of BRD by TUS can lead to a reduction in both the type and size of the lesion, leading to improved lung condition and tissue healing [27]. 

Furthermore, the difference between the two groups of the total hepatization areas was reduced from 8.05 cm^2^ at T0 to 2.50 cm^2^ at T5. According to the literature, lung consolidation refers to non-aerated lung areas and is associated with pneumonia during BRD [26,30,31]. The lung consolidation areas detected with TUS at a single moment may represent an acute or chronic inflammatory process [31]. The reduction in the size of the lesions is probably due to recovery processes of the lung tissue, and the reduction in the inflammatory process response due to early antibiotic and anti-inflammatory administration with the resolution of acute pneumonia caused by BRD. 

### 4.2. Clinical Follow-Up in Animals Affected by BRD

Periodic follow-up of animals with BRD may be useful in assessing both the response to medical treatments and the improvement of ultrasound scores of the lungs [27]. In the present study, the follow-up of group D showed an improvement of US with a difference between T0 and T1 (+1.5 days) that continued until the end of the study. Although the follow-up of RS was significant between T0 and T3, the PDA showed a difference between T1 and T3–T4 and between T2 and T4. According to Nutsch et al. (2005) and Torres et al. (2013), tulathromycin shows greater clinical efficacy than other antimicrobials for the treatment of BRD. In this study, group D showed an improvement in RS within 7–14 days from T1, similar to the results obtained after amoxicillin/clavulanic acid administration in the study by Hussein et al. (2018). Nevertheless, the weak relationship between RS and US [27] should be considered. In fact, despite the improvement in US after medical treatment (T0–T5), RS did not show differences between T0, T3, T4, and T5 in diseased animals, confirming the low sensitivity and specificity of the clinical examination [19]. Regarding nasal and ocular discharge, they showed improvement at T4 (+14 days) and T2 (+3 days) compared to T0, respectively. Hussein et al. (2018) showed an improvement in nasal discharge after 3 days of treatment. This difference in clinical response time may be due to a different medical protocol (amoxicillin/clavulanic acid versus tulathromycin).

In this study, the combination of tulathromycin and ketoprofen successfully reduced rectal temperature at T1. Thermography temperature showed a similar trend to rectal temperature, although the correlation between them was only 38.3%. Moreover, macrolides, especially tulathromycin, show anti-inflammatory and immunomodulatory effects [32,33,34]. The reduction in rectal temperature after 1.5 days from treatment may be related to a combined antibiotic and anti-inflammatory-mediated action [35]. However, the treatment of BRD should target both pathogens and the host inflammatory response to avoid uncontrolled self-perpetuating inflammation [34]. 

TUS is a diagnostic tool that can be used to assess the extent of lung lesions with varying degrees of tissue consolidation, inflammation, and fluid exudate deposits [35]. An increase in lung consolidation is associated with a reduction in average daily gain (ADG) with a consequent reduction in expectation growth [36,37]. An improvement in weight gain may be observed in animals affected by BRD treated with tulathromycin. The bioavailability following a single injection of this drug is more than 90%, with a mean concentration in lung tissue of 3.2, 4.1, 3.0, 1.9, and 1.2 µg/g after 0.5, 1, 7, 10, and 15 days, respectively. Furthermore, it must be considered that its typical MIC_90_ concentrations for *Pasteurella multocida* and *Mannheimia haemolytica* are 1 and 2 µg/mL, respectively [34]. 

In this study, the LS of the middle areas of both lung sides showed improvement during the follow-up in group D: the lung affected by hepatizations or fluid alveolograms showed healing to normal tissue at T3 (+7 days), T4 (+14 days), and T5 (+21 days). The LS of the cranial left lung showed an improvement in lung alteration with a reduction in the comet tail from 7 days after treatment until the end of the study. The sum of these improvements contributed to the reduction in the GL as of 3 days after treatment. These findings suggest that the general health status of lungs showed an improvement at 3 days after tulathromycin and ketoprofen treatment, although the individual health status of the affected areas of the lung showed the same improvement 7 days after treatment. 

The hepatization areas showed a similar trend over time. In fact, the mean value at T0 differed from T2 (only for the middle area of the left lung), T3, T4 (for the middle area of both sides), and T5 (for middle areas of both lung sides and the cranial area of the right lung). The total of hepatization areas showed a reduction from T0 to T1, T2, T3, T4, and T5, with a difference of about 6.00 cm^2^ after 7 days until the end of study. However, the persistence of hepatization areas at T5 may indicate the presence of chronic lesions. In fact, the use of antimicrobics is most effective in the early phase of the disease, before the onset of chronic and severe lesions [35]. In this case, the animals were not evaluated by TUS before their arrival at the restocking farm. The fluid alveolograms areas in the middle left lung differed from T0 to T3, while in the cranial left lung and middle right lung an improvement was reported from T0 to T4. However, the total of fluid alveologram areas showed a difference from T0 to T2 with a reduction of about 1.50 cm^2^ in the total area. 

All these findings suggest that the combination of tulathromycin and ketoprofen can be successfully used to treat the most affected areas of the lung during BRD [27,30,35], with an improvement of general lung’s health from 1.5 to 3 days after treatment. The monitoring and the positive development of these parameters is important for farm management and animal health. TUS evaluations of the different lung areas affected over the course of BRD show the same diagnostic accuracy; moreover, their regression appears to be highly sensitive in predicting the resolution of the inflammatory and infectious process [36,38]. The use of TUS as a non-invasive and feasible tool to monitor the efficacy of medical treatment during periodic follow-up [27] and to perform early diagnosis of BRD is indicated to improve animal welfare and reduce economic losses.

## 5. Conclusions

In this study, the use of TUS on fattening bulls on arrival at the farm was recognized as a useful diagnostic tool for screening for respiratory diseases. Indeed, the subjects identified as healthy did not present alterations compatible with BRD in contrast to the affected and treated group. Moreover, tulathromycin and ketoprofen treatment was effective in reducing US and rectal temperature after 1.5 days. Nasal and ocular discharges were reduced at 14 days and 3 days after treatment, respectively. Concerning lung lesions, the study confirmed that the cranial and middle areas on both sides of the lung were the most affected by BRD and showed an improvement in general health status as early as 3 days after treatment. In contrast, total areas of hepatization and fluid alveolograms were significantly reduced 1.5 and 3 days after treatment. At the end of the study, some of the areas of pulmonary hepatization remained, confirming that the treatment is more effective in acute conditions of the disease. 

## Figures and Tables

**Figure 1 animals-12-00994-f001:**
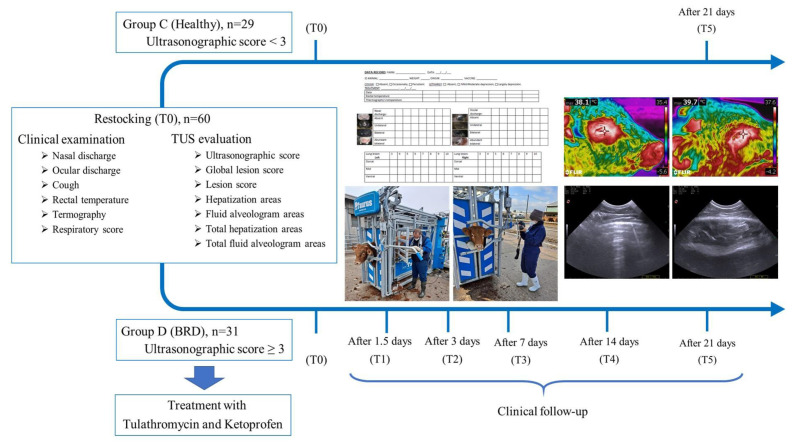
Experimental design flowchart of the study.

**Table 1 animals-12-00994-t001:** Comparison of parameters between group C and group D at T0 (Time 0; day of restocking) and T5 (Time 5; after 21 days). Data are expressed as median (1st quartile–3rd quartile; minimum value–maximum value) for not normally distributed data and as mean ± SEM for normally distributed data.

Parameters	TIME 0			TIME 5		
	Group C (n = 29)	Group D (n = 31)	*p*	Group C (n = 29)	Group D (n = 31)	*p*
US ^1^	2 (2–2; 2–2)	3 (3–4; 3–5)	<0.001	2 (2–2; 0–2)	2 (2–2; 0–3)	1.00
RS ^2^	3 (2.75–5; 1–7)	5 (4.5–6; 3–7)	<0.001	3 (3–5; 2–6)	5 (3.75–5.25; 2–7)	0.01
PDA ^3^	42.86%	83.87%	0.001	40.00%	75.00%	0.02
Nasal discharge	0 (0–2; 0–2)	2 (2–2; 0–2)	<0.001	1 (0–2;0–2)	2 (1.5–2; 0–3)	0.09
Ocular discharge	0 (0–0; 0–2)	0 (0–1; 0–2)	0.04	0 (0–0; 0–1)	0 (0–0; 0–2)	0.07
RT ^4^ (°C)	39.45 ± 0.46	40.07 ± 0.08	<0.001	39.48 ± 0.60	39.73 ± 0.44	0.09
TT ^5^ (°C)	38.03 ± 0.83	38.51 ± 0.26	0.08	39.74 ± 0.50	39.95 ± 0.54	0.15

^1^ Ultrasonography score; ^2^ Respiratory score; ^3^ Percentage of diseased animals; ^4^ Rectal temperature; ^5^ Thermography’s temperature.

**Table 2 animals-12-00994-t002:** Comparison of lung evaluations between group C and group D at T0 (Time 0; day of restocking) and T5 (Time 5; after 21 days). Data are expressed as median (1st quartile–3rd quartile; minimum value–maximum value) for not normally distributed data and as mean ± SEM for normally distributed data. Hepatization and fluid alveologram areas are expressed in cm^2^.

Parameters	TIME 0			TIME 5		
	Group C (n = 29)	Group D (n = 31)	*p*	Group C (n = 29)	Group D (n = 31)	*p*
Global lesion score	5 (4–7; 2–11)	12.5 (10–16; 6–24)	<0.001	4 (4–8; 0–15)	4 (2–8; 0–16)	0.86
Total of hepatizations areas	3.23 ± 0.65	11.29 ± 0.70	<0.001	2.66 ± 0.69	5.16 ± 0.69	0.09
Total of fluid alveolograms	0.02 ± 0.02	1.98 ± 0.35	0.01	0.03 ± 0.03	0.19 ± 0.10	0.62
**Caudal left lung—lesion score**	0 (0–0; 0–2)	0 (0–0; 0–6)	0.48	0 (0–0; 0–1)	0 (0–0; 0–0)	0.34
Hepatization	0.04 ± 0.23	0.20 ± 0.68	0.31	0 ± 0	0 ± 0	n.e. ^1^
Fluid alveolograms	0 ± 0	0.05 ± 0.24	0.33	0 ± 0	0 ± 0	n.e. ^1^
**Middle left lung—lesion score**	0 (0–2; 0–4)	3 (0.75–6; 0–7)	<0.001	0 (0–0; 0–4)	0 (0–1; 0–6)	0.23
Hepatization	0.48 ±0.87	2.18 ± 2.04	<0.001	0.15 ± 0.52	0.79 ± 1.41	0.05
Fluid alveolograms	0 ± 0	0.40 ± 0.80	0.02	0 ± 0	0.02 ± 0.09	0.33
**Cranial left lung—lesion score**	2 (0–2; 0–6)	4 (1.75–6; 0–7)	0.01	2 (0–2; 0–7)	2 (0–4; 0–6)	0.80
Hepatization	1.28 ± 1.24	2.51 ± 1.84	0.01	1.18 ± 0.94	1.72 ± 1.79	0.21
Fluid alveolograms	0.01 ± 0.05	0.41 ± 0.66	0.01	0.03 ± 0.13	0.07 ± 0.23	0.48
**Caudal right lung—lesion score**	0 (0–0; 0–4)	0 (0–0; 0–6)	0.91	0 (0–0; 0–0)	0 (0–0; 0–4)	0.34
Hepatization	0.04 ± 0.23	0.08 ± 0.42	0.67	0 ± 0	0.1 ± 0.49	0.33
Fluid alveolograms	0 ± 0	0.03 ± 0.16	0.33	0 ± 0	0 ± 0	n.e. ^1^
**Middle right lung—lesion score**	0 (0–1; 0–2)	2 (0–4.5; 0–6)	0.001	0 (0–1; 0–4)	0 (0–0.5; 0–4)	0.81
Hepatization	0.24 ± 0.77	2.28 ± 2.52	<0.001	0.4 ± 0.87	0.39 ± 1.01	0.97
Fluid alveolograms	0 ± 0	0.30 ± 0.74	0.05	0 ± 0	0 ± 0	n.e. ^1^
**Cranial right lung—lesion score**	2 (1–4; 0–6)	2 (2–6; 0–7)	0.05	2 (0.5–4; 0–7)	2 (0.5–3; 0–6)	0.55
Hepatization	1.27 ± 1.14	3.74 ± 2.15	<0.001	0.96 ± 0.83	2 ± 1.82	0.02
Fluid alveolograms	0.02 ± 0.09	0.47 ± 0.87	0.02	0.02 ± 0.08	0.1 ± 0.47	0.43

^1^ Not estimable.

**Table 3 animals-12-00994-t003:** Parameters’ trend of group D (n = 31) at T0 (Time 0; day of restocking), T1 (Time 1; after 1.5 days), T2 (Time 2; after 3 days), T3 (Time 3; after 7 days), T4 (Time 4; after 14 days), and T5 (Time 5; after 21 days). Data are expressed as median (1st quartile–3rd quartile; minimum value–maximum value) and mean ± SEM.

Parameters	T0	T1	T2	T3	T4	T5	*p*
US ^1^	3 ^a^ (3–4; 3–5)	2 ^b^ (2–3; 2–5)	2 ^c^ (2–2; 2–3)	2 ^c^ (2–2; 0–3)	2 ^c^ (2–2; 0–3)	2 ^c^ (2–2; 0–3)	<0.001
RS ^2^	5 ^a^ (4.5–6; 3–7)	5 ^a^ (4–5; 3–6)	5 ^a^ (4–5; 2–5)	4 ^b^ (2.25–5; 1–6)	3.5 ^b^ (3–5; 2–5)	5 ^a^ (3.75–5.25; 2–7)	<0.001
PDA ^3^	83.87% ^abc^	96.67% ^c^	87.10% ^bc^	63.33% ^ab^	46.15% ^a^	75.00% ^abc^	<0.001
Nasal discharge	2 ^ab^ (2–2; 0–2)	2 ^b^ (2–2; 2–3)	2 ^ab^ (2–2; 0–2)	2 ^ac^ (0–2; 0–2)	1.5 ^c^ (0–2; 0–2)	2 ^abc^ (1.5–2; 0–3)	<0.001
Ocular discharge	0 ^a^ (0–1; 0–2)	0 ^ab^ (0–0; 0–2)	0 ^b^ (0–0; 0–1)	0 ^b^ (0–0; 0–1)	0 ^b^ (0–0; 0–1)	0 ^ab^ (0–0; 0–2)	0.001
RT ^4^ (°C)	40.07 ± 0.08 ^a^	39.36 ± 0.08 ^b^	39.40 ± 0.08 ^b^	39.46 ± 0.09 ^b^	39.43 ± 0.09 ^b^	39.74 ± 0.10 ^ab^	<0.001
TT ^5^ (°C)	38.39 ± 0.29 ^b^	36.42 ± 0.23 ^c^	36.33 ± 0.23 ^c^	35.95 ± 0.23 ^c^	37.08 ± 0.26 ^c^	39.97 ± 0.26 ^a^	<0.001

^1^ Ultrasonography score; ^2^ Respiratory score; ^3^ Percentage of diseased animals; ^4^ Rectal temperature; ^5^ Thermography’s temperature; ^a–c^ Mean values in the same row which differ significantly (*p* ≤ 0.05).

**Table 4 animals-12-00994-t004:** Lung evaluations’ trend of group D (n = 31) at T0 (Time 0; day of restocking), T1 (Time 1; after 1.5 days), T2 (Time 2; after 3 days), T3 (Time 3; after 7 days), T4 (Time 4; after 14 days), and T5 (Time 5; after 21 days). Data are expressed as median (1st quartile–3rd quartile; minimum value–maximum value) and mean ± SEM. Hepatization and fluid alveologram areas are expressed in cm^2^.

Parameters	T0	T1	T2	T3	T4	T5	*p*
Global lesion score	12.5 ^a^ (10–16; 6–24)	10 ^ab^ (8–13; 4–24)	8 ^bc^ (4–12.5; 2–19)	4 ^c^ (4–8; 0–19)	4 ^c^ (4–8; 0–15)	4 ^c^ (2–8; 0–16)	<0.001
Total of hepatizations areas	11.06 ± 0.75 ^a^	8.08 ± 0.66 ^b^	6.68 ± 0.65 ^bc^	5.14 ± 0.66 ^c^	4.62 ± 0.70 ^c^	5.10 ± 0.75 ^c^	<0.001
Total of fluid alveolograms	1.74 ± 0.44 ^a^	0.95 ± 0.25 ^ab^	0.27 ± 0.13 ^b^	0.31 ± 0.14 ^b^	0.18 ± 0.11 ^b^	0.21 ± 0.14 ^b^	<0.001
**Caudal left lung—lesion score**	0 (0–0; 0–6)	0 (0–0; 0–2)	0 (0–0; 0–1)	0 (0–0; 0–0)	0 (0–0; 0–4)	0 (0–0; 0–0)	0.23
Hepatization	0.20 ± 0.68	0.09 ± 0.48	0 ± 0	0 ± 0	0.07 ± 0.35	0 ± 0	0.29
Fluid alveolograms	0.05 ± 0.24	0 ± 0	0 ± 0	0 ± 0	0 ± 0	0 ± 0	0.30
**Middle left lung—lesion score**	3 ^a^ (0.75–6; 0–7)	1 ^ab^ (0–2; 0–6)	0 ^abc^ (0–2; 0–7)	0 ^c^ (0–0; 0–6)	0 ^c^ (0–0; 0–2)	0 ^bc^ (0–1; 0–6)	<0.001
Hepatization	2.18 ± 0.04 ^a^	1.14 ± 1.41 ^ab^	0.81 ± 1.17 ^b^	0.45 ± 1.22 ^b^	0.31 ± 0.90 ^b^	0.79 ± 1.41 ^b^	<0.001
Fluid alveolograms	0.40 ± 0.80 ^a^	0.19 ± 0.58 ^ab^	0.04 ± 0.11 ^ab^	0.03 ± 0.10 ^b^	0 ± 0 ^b^	0.02 ± 0.09 ^b^	0.02
**Cranial left lung—lesion score**	4 ^a^ (1.75–6; 0–7)	4 ^a^ (2–6; 0–7)	2 ^ab^ (2–5; 0–7)	2 ^b^ (0–3.75; 0–7)	2 ^b^ (0–4; 0–7)	2 ^b^ (0–4; 0–6)	0.03
Hepatization	2.51 ± 1.84	2.77 ± 1.96	2.32 ± 1.50	1.97 ± 1.67	1.73 ± 1.64	1.72 ± 1.79	0.18
Fluid alveolograms	0.41 ± 0.66 ^a^	0.30 ± 0.58 ^ab^	0.13 ± 0.29 ^ab^	0.19 ± 0.49 ^ab^	0.04 ± 0.13 ^b^	0.07 ± 0.23 ^b^	0.03
**Caudal right lung—lesion score**	0 (0–0; 0–6)	0 (0–0; 0–0)	0 (0–0; 0–0)	0 (0–0; 0–0)	0 (0–0; 0–0)	0 (0–0; 0–4)	0.42
Hepatization	0.09 ± 0.42	0 ± 0	0 ± 0	0 ± 0	0 ± 0	0.10 ± 0.49	0.41
Fluid alveolograms	0.03 ± 0.16	0 ± 0	0 ± 0	0 ± 0	0 ± 0	0 ± 0	0.32
**Middle right lung—lesion score**	2 ^a^ (0–4.5; 0–6)	2 ^ab^ (0–3; 0–7)	0 ^abc^ (0–2; 0–7)	0 ^bc^ (0–0.75; 0–6)	0 ^c^ (0–0; 0–4)	0 ^bc^ (0–0.5; 0–4)	<0.001
Hepatization	2.28 ± 2.52 ^a^	1.51 ± 1.72 ^ab^	1.03 ± 1.61 ^abc^	0.63 ± 1.23 ^bc^	0.28 ± 0.82 ^c^	0.39 ± 1.01 ^bc^	<0.001
Fluid alveolograms	0.30 ± 0.74 ^a^	0.17 ± 0.54 ^ab^	0.03 ± 0.11 ^ab^	0.02 ± 0.07 ^ab^	0 ± 0 ^b^	0 ± 0 ^b^	0.02
**Cranial right lung—lesion score**	2 (2–6; 0–7)	2 (2–6; 0–7)	2 (2–4; 0–7)	2 (2–4; 0–7)	2 (1–4; 0–7)	2 (0.5–3; 0–6)	0.25
Hepatization	3.74 ± 2.15 ^a^	2.65 ± 1.85 ^ab^	2.53 ± 1.45 ^ab^	2.09 ± 1.42 ^b^	2.29 ± 1.86 ^ab^	2.00 ± 1.82 ^b^	0.02
Fluid alveolograms	0.47 ± 0.87	0.37 ± 0.74	0.09 ± 0.23	0.09 ± 0.31	0.15 ± 0.37	0.10 ± 0.47	0.15

^a–c^ Mean values in the same row which differ significantly (*p* ≤ 0.05).

## Data Availability

The data are available by sending an email to the corresponding author.

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
