# Peer review of "Lung Ultrasonography and Clinical Follow-Up Evaluations in Fattening Bulls Affected by Bovine Respiratory Disease (BRD) during the Restocking Period and after Tulathromycin and Ketoprofen Treatment"

_animals, 2022, doi:10.3390/ani12080994_

Round 1
Reviewer 1 Report
The manuscript assesses the capacity of ultrasonography as a means of screening and evaluates the effectiveness of a drug product in controlling pulmonary lesions caused by BRD. The manuscript is well organized and schematic, with interesting evaluation points for the management of fattening bull’s livestock.
In my opinion, the manuscript could be accepted after minor revisions:
- Line 17, you should add the simple summary
- Line 83: Change in “This study was a case time-series…”
- You have written about a deep nasal swab (lines 133-138) then you should add information about its result in Results section (line 177)
- You should delete “-value” from “p-value” in the text and tables
- In Discussion section you should delete every reference to “significant” or “significantly” because you discuss only significant data.
- Lines 265-268, you should also consider that the McGuirk and Peek score was created on calves, could this contribute give you a low significance? Please, add this information in the text.
Author Response
Reviewer 1
The manuscript assesses the capacity of ultrasonography as a means of screening and evaluates the effectiveness of a drug product in controlling pulmonary lesions caused by BRD. The manuscript is well organized and schematic, with interesting evaluation points for the management of fattening bull’s livestock.
In my opinion, the manuscript could be accepted after minor revisions:
1) Line 17, you should add the simple summary.
AU: We would like to apologize for this mistake in submission. We have included the simple summary in Line 17-25:
“Stressors as transport may predispose animals to bovine respiratory disease due both to viral and bacterial agents. This syndrome reduces animal well-being and increase economic losses, commonly because of the challenging diagnosis. The ultrasonography is a non-invasive, real-time, and portable method to investigate frequently employed in farm practice. The aim of this study was to evaluate the effectiveness of tulathromycin and ketoprofen treatment in animals with bovine respiratory syndrome at the arrive to the restocking farm using thoracic ultrasonography. Our findings reveals that the lung ultrasound is a useful diagnostic tool for screening the respiratory diseases. Moreover, tulathromycin and ketoprofen treatment was effective in reduce both clinical symptoms and lung’ lesions”.
2) Line 83: Change in “This study was a case time-series…”
AU: Line 103, we added “This study was a case time-series analysis”.
3) You have written about a deep nasal swab (lines 133-138) then you should add information about its result in Results section (line 177)
AU: Lines 221-224, we added “Animals affected by BRD (group D) were positive for Pasteurella multocida and Mannheimia haemolytica on deep nasal swab. A single pool carried out from three animals showed a weak reactivity for Mycoplasma bovis. All these pathogens were sensitive to the tulathromycin used for their treatment based on MIC analysis”.
4) You should delete “-value” from “p-value” in the text and tables.
AU: We deleted “-value” from “p-value” in the manuscript.
5) In Discussion section you should delete every reference to “significant” or “significantly” because you discuss only significant data.
AU: We have deleted the required reference in the “Discussion” paragraph and made some English corrections in the manuscript as suggested by reviewer 2 and 3.
6) Lines 265-268, you should also consider that the McGuirk and Peek score was created on calves, could this contribute give you a low significance? Please, add this information in the text.
AU: Lines 332-335, we added “Nevertheless, the RS was originally designed for the classification of calves: the inaccuracy in detection of respiratory disease in beef fattening bulls is probably due to the greater variety of clinical presentation and severity of BRD in adult animals and their behavioral response to stressors”.
Reviewer 2 Report
The paper needs a lot of editing and correcting of grammar with simple mistakes throughout eg Spelling, New Zeland, (New Zealand), Ultrasonograhyc (ultrasonography).
The study seems illogical
Why is it group C and D? What happened to group A and B?
Why was ultrasound not undertaken in group C at the same time points as group D?
Swabs for microbiology culture were collected from group D. I could not find the results in the paper.
Author Response
Dear Editor and Reviewers,
All authors would like to thank you for reviewing this manuscript.
We would like to thank Reviewers for the suggestions they have provided us. In accordance with the Reviewers' suggestions, the manuscript was revised and corrected for English grammar and syntax.
The authors partitioned and numbered Reviewers' commentary for ease of response and reading. Authors' responses are preceded by the abbreviation AU and are colored blue.
In the manuscript, the “Track changes” function of Microsoft Word was used to evidence changes as suggested in the email of the Editor.
All authors meet the Journal's criteria for authorship.
Best Regards
Enrico Fiore
----------------
Reviewer 2
1) The paper needs a lot of editing and correcting of grammar with simple mistakes throughout eg Spelling, New Zeland, (New Zealand), Ultrasonograhyc (ultrasonography).
AU: We asked to a native speaker to perform English corrections.
-Table 1 and 3, we changed “ultrasonograhyc score” in “ultrasonography score”.
Regarding spelling, we followed the directions in the template provided on the journal's website in which the spelling "English (United States)" was used.
2) The study seems illogical. Why is it group C and D? What happened to group A and B?
Why was ultrasound not undertaken in group C at the same time points as group D?
AU: We checked the text to be sure, but Groups A and B were never named. In fact, we named groups C and D as "Control" and "Disease" groups. Regarding the number and timing of ultrasounds examinations, we performed more evaluations in group D than in group C because we wanted to assess if and in what time frame a healing or improvement in lung condition was realized following the treatment. This choice was made to provide a guide to veterinarians on when to perform a follow-up on treated animals to understand if the treatment is having an effect or not.
3) Swabs for microbiology culture were collected from group D. I could not find the results in the paper.
AU: As also mentioned in reviewer 1's comment 3, we added in Lines 221-224 “Animals affected by BRD (group D) were positive for Pasteurella multocida and Mannheimia haemolytica on deep nasal swab. A single pool carried out from three animals showed a weak reactivity for Mycoplasma bovis. All these pathogens were sensitive to the tulathromycin used for their treatment based on MIC analysis”.
Reviewer 3 Report
The subject tackled by the authors in this manuscript is of both health and economic interest. They propose a very advantageous and low-cost method for diagnosing and follow-up of BRD with statistically supported success. Nevertheless, the design has some gaps, at least not well motivated. The discussion could be more correlative, rather than descriptive.
Please find further comments in the attached document.

Author Response
Cover letter
Dear Editor and Reviewers,
All authors would like to thank you for reviewing this manuscript.
We would like to thank Reviewers for the suggestions they have provided us. In accordance with the Reviewers' suggestions, the manuscript was revised and corrected for English grammar and syntax.
The authors partitioned and numbered Reviewers' commentary for ease of response and reading. Authors' responses are preceded by the abbreviation AU and are colored blue.
In the manuscript, the “Track changes” function of Microsoft Word was used to evidence changes as suggested in the email of the Editor.
All authors meet the Journal's criteria for authorship.
Best Regards
Enrico Fiore
-------------
Reviewer 3
The subject tackled by the authors in this manuscript is of both health and economic interest. They propose a very advantageous and low-cost method for diagnosing and follow-up of BRD with statistically supported success. Nevertheless, the design has some gaps, at least not well motivated. The discussion could be more correlative, rather than descriptive.
Please find further comments in the attached document.
1) The Simple abstract is still missing at my profile link!
AU: We would like to apologize for this mistake in submission. We have included the simple summary in Line 17-25:
“Stressors as transport may predispose animals to bovine respiratory disease due both to viral and bacterial agents. This syndrome reduces animal well-being and increase economic losses, commonly because of the challenging diagnosis. The ultrasonography is a non-invasive, real-time, and portable method to investigate frequently employed in farm practice. The aim of this study was to evaluate the effectiveness of tulathromycin and ketoprofen treatment in animals with bovine respiratory syndrome at the arrive to the restocking farm using thoracic ultrasonography. Our findings reveals that the lung ultrasound is a useful diagnostic tool for screening the respiratory diseases. Moreover, tulathromycin and ketoprofen treatment was effective in reduce both clinical symptoms and lung’ lesions”.
2) Introduction: Bacterial infections should also be mentioned as evolving per primam, not only as secondary infections, and other potential agents included.
AU: Lines 53-55, we added “to Mannheimia haemolytica, Mycoplasma bovis, Pasteurella multocida, or Histophilus somni, Salmonella spp., or Actinomyces pyogenes. In addition, some of these bacteria can also cause a primary infection”.
3) Line 48, Please rephrase “Commonly, the clinical observations of them are used for diagnosis”.
AU: Lines 57-58, we changed “Commonly, the clinical observations of them are used for diagnosis”
In
“Commonly, the observations of these clinical signs are utilized for BRD diagnosis in-field”
4) Lines 58-59, Could you please explicit “Ultrasonography is a non-invasive, cost-effective and portable method to investigate 58 in real-time different structures”.
AU: Ultrasound is considered:
- non-invasive because it is necessary to simply place a probe on the animal to examine deep structures, rather than performing blood sampling or biopsy;
- cost-effective because now all veterinarians have an ultrasound scanner with at least 1 probe that can be used to evaluate animals;
- portable because ultrasound systems have also been developed for veterinary medicine that allow an used in field;
- real-time investigation because we can see a different tissue conditions indicating for example inflammation in the same moment of ultrasound examination, rather than waiting for laboratory reports.
5) Lines 63-64, Please rephrase unclear “These findings are 63 important to reduce animal suffering and avoid unnecessary antimicrobial treatment 64 for reducing the antimicrobial resistance”.
AU: Lines 77-81, we changes “These findings are important to reduce animal suffering and avoid unnecessary antimicrobial treatment for reducing the antimicrobial resistance”
In
“The greater sensitivity of ultrasonography is important for an early detection of diseased animals therefore reducing animal suffering and chronicization of the disease, whereas the greater specificity is important for accurate evaluation of the animals’ health status, in order to avoid unnecessary antimicrobial treatment, for reducing the antimicrobial resistance risk development”
6) Lines 68-70 If the description of the antibiotic includes the group, maybe it should be compared to other groups and their representatives? “Tulathromycin is a macrolide that showed lower events of morbidity, mortality and retreatment when compared to others such as tilmicosin, oxytetracycline, florfenicol, ceftiofur, and gamithromycin”.
AU: Lines 87-89, we changed “to others such as tilmicosin, oxytetracycline, florfenicol, ceftiofur, and gamithromycin”
In
“to other macrolides (tilmicosin, and gamithromycin), tetracyclines (oxytetracycline), amphenicols (florfenicol), and cephalosporin (ceftiofur)”.
7) Lines 89-90 The authors mention “All animals were weighed and vaccinated against bovine Parainfluenza-3 Virus (PI3), 88 Bovine Viral Diarrhea Virus (BVDV), Bovine Herpes Virus-1 (BHV-1) and Bovine Respir- 89 atory Syncytial Virus (BRSV) (Cattle Master 4, Zoetis Italia S.r.l., Rome, Italy) on the day 90 of restocking and after 21 days” and also “On the day of restocking, animals were divided into two groups according to ultra- 122 sonographic score: group C and group D. The group C or control/healthy group had an 123 ultrasonographic score lesser than 3 and enrolled 29 fattening bulls. Group D or disease 124 group had an ultrasonographic score greater or equal to 3 and enrolled 31 fattening bulls”. It is unclear if the animals were vaccinated first and the same day groupd in C and D, or some time elapsed between the two operations?
AU: In the Italian restocking method, fattening bulls are weighed and vaccinated on the day of restocking at the farm. To do this, all animals are led into a runway that ends with a cattle crush (lines 113-114; Fig.1). All operations such as clinical examination, ultrasound evaluation, weighing and vaccination were performed while the animals were individually blocked in the cattle crush on their day of restocking (T0). The same procedure was applied after 21 days (T5).
According to the ultrasonography score based on Ollivett and Buczinski, 2016, animals were divided into two groups. The animals identified as affected by BRD (group D) received the deep nasal swab and were treated. The animals were released from cattle crush only after all these procedures were completed.
8) Fig. 1. Page 4 Was the injection administered to group D on day 0, of the first examination and
decision on the groups? It is not clear from the fig. at what interval it was performed, except it was
subsequent to division, in group D. In case it happened on the day of restocking, it was performed
before or after the vaccination of the animals?
AU: Figure 1 showed that at restocking (T0), all animals received a clinical examination and a ultrasonographic evaluation. Based on the described criteria they were divided into the two groups, and group D was treated as also specified in the previous commentary. Then, the figure indicates in what timing the animals were re-evaluated according to the group and showed some figures: the individual data record, the animal inside the cattle crush during the ultrasound evaluation and the evaluation with the thermal camera, the thermographic and ultrasound images.
Vaccination and weighing of animals were the first two procedures performed when animals were in the cattle crush due to husbandry practices.
9) Line 129-130 “No one animal of group C showed signs indicative of BRD or presented an ultrasonographic score greater or equal to 3 during the study”, being considered healthy. In table 1, page 5, under PDA (percentage of diseased animals) for group C was 42.86%; who were this animals in the healthy group? Those that had a positive <3 lung ultrasonography? This is a quite high subclinical percentage compared to 83.87% with clinical and lung ultrasonography signs? Why weren’t they monitored then at all 5 moments in time? Similarly, the decrease from moment 0 to moment 5 was quite low in both groups 40.00 % versus 75.00 % (2.86% versus 8.87%) following the treatment. °C should be indicated in the table where it is the case.
AU: Animals found to be diseased by respiratory score (PDA) all had an ultrasound score less than or equal to 2. The issue was the rectal temperature. In fact, the respiratory score provides a score of 3 if the temperature is greater than or equal to 39.4 °C. So, if an animal had this temperature and a slight unilateral ocular and nasal discharge, or a mild ocular/nasal discharge, to fall into the sick category. As also stated in the discussion (lines 332-335), the respiratory score was created based on calves that are much more accustomed to handling than fattening bulls. In fact, these animals may exhibit hyperthermia due to stress generated by transport or manipulation by handlers. Therefore, the respiratory score and the PDA was in our opinion biased for the evaluation of this category of animals. In addition, all animals were taken to the runway at each evaluation time, so even healthy animals were quickly reevaluated for temperature and discharge, and no animal ever showed signs suggesting BRD.
In table 1 and 3 we added “ (°C) ” in both RT and TT parameters.
10) Table 3. How do the authors explain the sharp increase of PDA (75.00%) at T5?
AU: The absolute value of the PDA at T5 is greater than at T3 and T4, but the statistical test used (k proportions chi-squares test) did not evidences a significant difference between T5 and other time points.
11) Lines 133-136 “Animals in group D received a deep nasal swab before drug treatment (35 cm, equipped with sheath and placed in an agarized blister pack; Medical Wire & Equipment Co Ltd, Corsham, United Kingdom). The nasal swabs were refrigerated in a cold box at 4 °C and transported at the same temperature to the laboratory of IZSVe within 1 hour. They were used for a bacteriological
examination and determination of Minimum Inhibitory Concentration (MIC) according to the validated protocol of the IZSVe”. Where were the results of these tests described in the Results section?
AU: As also suggested by reviewer 1’ comment 3, we added in Lines 221-224 “Animals affected by BRD (group D) were positive for Pasteurella multocida and Mannheimia haemolytica on deep nasal swab. A single pool carried out from three animals showed a weak reactivity for Mycoplasma bovis. All these pathogens were sensitive to the tulathromycin used for their treatment based on MIC analysis”.
Round 2
Reviewer 2 Report
Please review the grammar and spelling again.
The simple summary needs major grammatical revision.
Spell check the entire paper
eg Line 101 New Zealand is still spelt wrongly ("New Zeland"
Author Response
Reviewer 2
1) Please review the grammar and spelling again. The simple summary needs major grammatical revision. Spell check the entire paper eg Line 101 New Zealand is still spelt wrongly ("New Zeland").
AU: We checked the manuscript for further corrections to the English spelling.
Section Managing Editor
AU: As requested by the Section Managing Editor, we added the “Institutional Review Board Statement”, moving the sentences from Materials and Methods to this section.
“No approval from Ethics Committee was required. No invasive medical procedures were executed to perform the study. The study was performed with the consent of the animals’ owner during the routinary clinical activity of the Veterinary Teaching Hospital, University of Padua. Animal care and procedures are in accordance with the Guide for the Care and Use of Laboratory Animals and Directive 2010/63/EU for animal experiments (National law: D.L. 26/2014).”
